# Reverso: Efficient Time Series Foundation Models for Zero-shot Forecasting

**Xinghong Fu** [1]   **Yanhong Li** [2]   **Georgios Papaioannou** [3]   **Yoon Kim** [1]

Code: https://github.com/shinfxh/reverso

## Abstract

Learning time series foundation models has been shown to be a promising approach for zero-shot time series forecasting across diverse time series domains. Insofar as scaling has been a critical driver of performance of foundation models in other modalities such as language and vision, much recent work on time series foundation modeling has focused on scaling. This has resulted in time series foundation models with hundreds of millions of parameters that are, while performant, challenging to deploy, especially in memory-constrained settings such as in edge and on-device deployment. This paper describes a simple parameter-efficient architecture for time-series foundation modeling. Our architecture consists of three components: a multi-scale input strategy where we create multiple versions of the original input via downsampling at different scales; hybrid sequence-mixing layers consisting of long convolutions and DeltaNet layers; and an attention-based decoder head. We combine our architecture with standard data augmentation recipes to train a family of parameter-efficient time series foundation models, dubbed *Reverso*. Reverso achieves competitive performance across zero-shot forecasting, classification and anomaly detection, pushing the performance-efficiency Pareto frontier.

## 1. Introduction

Time series forecasting is a core problem in machine learning with widespread applications in weather forecasting, energy grid analysis, supply chain logistics, financial predictions, and more. Traditionally, statistical models (Box and Jenkins, 1970; Engle, 1982; Bollerslev, 1986; Harvey, 1990; Hyndman and Khandakar, 2008) as well as deep learning approaches based on RNNs (Elman, 1990; Hochreiter and Schmidhuber, 1997; Cho et al., 2014) have enjoyed great success in time series forecasting (Goel et al., 2017; Qin et al., 2017; Petneházi, 2019; Hewamalage et al., 2021, *i.a.*). More recently, models based on the transformer architecture (Vaswani et al., 2017) have led to further improvements (Nie et al., 2023; Zhou et al., 2021; Wu et al., 2021; Zhou et al., 2022; Liu et al., 2022, *i.a.*). These initial deep learning-based approaches to time series forecasting were dataset-specific, and thus trained models for particular domains/tasks of interest. While such models can attain high accuracy when sufficient in-distribution data are available, they incur substantial costs in data collection and model training/maintenance. This approach stands in stark contrast to recent progress in domains such as language, vision, and biology, where *foundation models* pretrained on broad datasets have been found to be useful across many tasks with little or no task-specific training (Bommasani, 2021).

The successes of foundation models in other modalities have motivated the recent line of work on *time series foundation models* (TSFM; Garza et al., 2023; Ansari et al., 2024; Das et al., 2024; Liu et al., 2024; Woo et al., 2024; Liu et al., 2025b;c; Graf et al., 2025; Auer et al., 2025b; Moroshan et al., 2025, *i.a.*). TSFMs are large-scale neural networks trained on heterogeneous time series data taken from broad domains (see Liang et al. (2024) and Kottapalli et al. (2025) for surveys). A particularly useful capability of decoder-based TSFMs is their ability to perform *zero-shot forecasting* via in-context learning, i.e., predicting the future given any historical time series data given as context. These TSFMs can thus serve as a domain-general tool for time series forecasting. However, insofar as scaling has been a critical driver of progress of foundation models in other domains, much existing work has focused on scaling TSFMs, i.e., training ever-larger models on ever-larger datasets. For example, Sun et al. (2025) train a series of models up to 1.5B parameters and observe continuous improvements with scaling model size. While such large models can be performant, their sheer size can make them prohibitively expensive to train and deploy. This is especially the case when considering on-device and edge deployment scenarios, where the

[1]Massachusetts Institute of Technology [2]Allen Institute for AI [3]Qube Research & Technologies. Correspondence to: Xinghong Fu <fxh@mit.edu>.

*Proceedings of the $2^{nd}$ ICML Workshop on Foundation Models for Structured Data*, Seoul, South Korea. 2026. Copyright 2026 by the author(s).

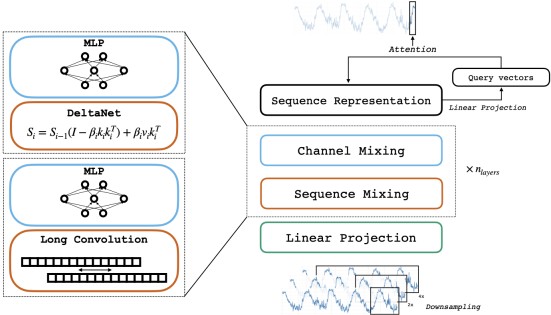

*Figure 1.* Reverso architecture. Given a sequence of length $S$, we turn it into a length-$L$ input $t \in \mathbb{R}^{L \times C}$ by taking the $L, 2L, \ldots, 2^{C-1}L$ most recent data points and downsampling by $\times 1, \times 2, \ldots, \times 2^{C-1}$. The input $t$ is passed through a single projection layer to obtain embedding representations $x \in \mathbb{R}^{L \times d}$. Then, $n_{layers}$ of sequence-mixing and channel-mixing blocks operates on $x$, where we alternate between long convolutions and DeltaNet for sequence mixing across length $L$, and use MLP layers for channel mixing across dimension $d$. The final output head (based on an attention-based transformation) obtains the predictions $\hat{y} \in \mathbb{R}^p$.

amount of memory available is orders of magnitude smaller than in cloud deployment.

Motivated by efficiency concerns, this work develops a parameter-efficient architecture for time-series foundation modeling. Our architecture is simple, and makes use of long convolution layers (Fu et al., 2023) and modern linear RNN layers (in particular DeltaNet layers (Schlag et al., 2021; Yang et al., 2024b)). We combine the hybrid architecture with an attention-based decoder head and a multi-scale input strategy where we feed multiple versions of the input via downsampling at different scales. This enables efficient modeling of longer-contexts without increasing the model's sequence length. We combine our architecture with standard data augmentation and inference-time strategies to arrive at a simple recipe that works well in practice. With our recipe, we train a family of TSFMs (dubbed *Reverso*) from 0.2M to 2.6M parameters that significantly push the performance-efficiency frontier. The parameter-efficiency of Reverso translates to inference-efficiency. Reverso attains excellent performance across zero-shot forecasting, classification, anomaly detection and imputation, thus positioning it as an attractive architecture for learning time-series foundation models.

## 2. Methods
Here we describe our recipe for learning parameter-efficient TSFMs, which includes the architecture (§2.1), dataset (§B.1), and inference strategy (§B.2).

### 2.1. Architecture
We are given an input time series $t_{in} \in \mathbb{R}^S$ of length $S$ and must predict an output $y \in \mathbb{R}^T$ of length $T$. Following standard practice (Nie et al., 2023), we parameterize our model to take in a fixed sequence of length $L$ and train it to predict a patch of $p$ points at a time (in parallel) through

learning a function $f_\theta : \mathbb{R}^L \to \mathbb{R}^p$ parameterized with $\theta$. During inference, we autoregressively predict chunks of $p$ data points until we have forecasted $T$ points. We use $L = 2048, p = 48$ for Reverso.

Our model architecture, shown in Figure 1, is extremely simple and consists of a multi-scale input that is fed to stacked neural network blocks where each block consists of a sequence mixing module followed by an MLP channel mixing module. Finally, we have an output decoder based on attention that uses the contextualized representation of the input to predict $p$ data points at once.

**Multi-scale input.** We first create a multi-scale input $t \in \mathbb{R}^{L \times C}$ by creating a downsampled version of the input. Concretely, a channel $c$ (where $1 \leq c \leq C$) takes the $2^{c-1} \times L$ most recent data points and downsamples with a stride of $2^{c-1}$. This has two effects: first, it allows the model to condition longer contexts without increasing the model input length $L$. Second, it forces the architecture to model time-series at multiple scales, which we hypothesize serves as a useful source of inductive bias. In practice we use $C = 4$, so the effective context length is 16384 and $t \in \mathbb{R}^{L \times 4}$.

**Normalization.** For each channel in $t \in \mathbb{R}^{L \times C}$, we normalize to range $[0, 1]$ by subtracting the $\min$ and dividing by $\max - \min$. We found this $[0, 1]$-normalization to work better than $z$-score normalization which subtracts the mean and divides by the standard deviation. We unnormalize the model's output for prediction. In cases where there are missing values, these are imputed using linear interpolation. For sequences shorter than the model context length $L$, the remaining values are back-filled using the leftmost available data point.

**Embedding.** The normalized sequence $t$ is then up-projected pointwise using a single linear layer into $d$ dimensions, yielding a transformed sequence $x \in \mathbb{R}^{L \times d}$. Unlike existing works that make use of special time embeddings (Moroshan et al., 2025; Alexandrov et al., 2019) to include seasonality and frequency features, utilizing metadata that might not be present at inference time, we adopt a minimalistic approach that can handle any time series as a simple numeric sequence.

**Sequence mixing.** We adopt a hybrid sequence mixing strategy wherein we switch between (gated) long convolution (Fu et al., 2023) and DeltaNet layers (Schlag et al., 2021; Yang et al., 2024b). The long convolution layer uses depthwise separable convolutions (Chollet, 2017), where the number of groups is equal to $d$. This obtains the output $z \in \mathbb{R}^{L \times d}$ from an input sequence $x \in \mathbb{R}^{L \times d}$ given convolution kernel weight $w \in \mathbb{R}^{k \times d}$ via $z_{i,j} = \sum_{m=0}^{k-1} w_{m,j} \cdot x_{i-m,j}$ where $0 \leq i \leq L - 1$ indexes the sequence position, and $0 \leq j \leq d - 1$ indexes the dimensions. The long convolu-

tion is an instance of the convolution kernel where $k = L$. We also make use of a gating layer, where the gate comes from a depthwise separable (short) convolution layer. Taken together, our convolutional sequence mixing primitive is given by

$$x_{conv} \leftarrow \text{SiLU}(\text{short-conv}(x) \odot \text{long-conv}(x)),$$
$$x \leftarrow x + \text{LayerNorm}(x_{conv}).$$

With FFT the overall complexity of the convolution layer is $O(dL \log L)$, enabling faster training than standard attention. While the FFT-based convolutions was previously not optimized for GPUs, recent works have enabled significant wallclock speed-ups (Fu et al., 2023), which we make use of in practice.

We also make use of DeltaNet (Schlag et al., 2021) every other layer, which learns the following state transition using query, key and value vectors $q_i, k_i, v_i \in \mathbb{R}^{d_h}$ (with head dimension $d_h$)

$$S_i = S_{i-1}(I - \beta_i k_i k_i^T) + \beta_i v_i k_i^T,$$
$$x_i \leftarrow x_i + \text{LayerNorm}(S_i q_i).$$

Here the query, key, value vectors are obtained from linear projections followed by short convolutions of the input $x$, and $\beta_i \in (0, 1)$ is obtained by a linear projection of the input $x_i$ followed by a sigmoid. We use 4 heads (i.e., $d_h = \frac{d}{4}$). To better model bidirectional context over the entire length $L$ sequence, we add the last time step of the previous layer to the current layer's first hidden state (i.e., $x_0^{(l)} \leftarrow x_0^{(l)} + x_{L-1}^{(l-1)}$) before the DeltaNet layer. We found this type of vector-based "state-weaving" strategy to work well in practice, similar to Moroshan et al. (2025).

In our ablation studies we also compare against other DeltaNet variants such as Gated DeltaNet (GDN; Yang et al., 2025) and Gated Delta Product (GDP; Siems et al., 2025), as well as linear attention variants such Gated Linear Attention (GLA; Yang et al., 2024a) (which generalizes state-space models such as Mamba-2 (Dao and Gu, 2024)). Our findings show that DeltaNet performs well despite having fewer parameters.

**Channel mixing.** Each sequence mixing layer is followed by a channel mixing MLP layer. The MLP layer works as in the standard transformer architecture (Vaswani et al., 2017), with a dimension expansion factor of 4, with ReLU activations. We found this simple MLP to work better than GLU-variants (Shazeer, 2020).

**Decoder head.** The above blocks transform a sequence of inputs $x^{(0)} \in \mathbb{R}^{L \times d}$ into $x^{(n)} \in \mathbb{R}^{L \times d}$ after $n$ layers. To obtain the final prediction, we first pass the final transformed input $x^{(n)}$ obtain a set of decoder "query" vectors $q_{dec}$ through a bilinear transformations:

$$q_{dec} = W_L x^{(n)} W_q, \quad W_L \in \mathbb{R}^{p \times L}, W_q \in \mathbb{R}^{d \times d}, q_{dec} \in \mathbb{R}^{p \times d}$$

The decoder query vectors are then used to attend over the keys and values,

$$k_{dec} = x^{(n)} W_k, \qquad v_{dec} = x^{(n)} W_v,$$
$$o = \text{attention}(q_{dec}, k_{dec}, v_{dec}).$$

We use `sin-cos` positional embeddings. Finally, we apply a linear layer to obtain the final output $\hat{y} \in \mathbb{R}^p$, $\hat{y} = o \, w_o, w_o \in \mathbb{R}^{d \times 1}$. We found this type of attention-based decoder "head" to be more performant than a simple linear layer that directly predicts a $p$-sized vector from $x^{(n)}$.

**Training objective.** Given the model prediction $\hat{y}$ we unnormalized the output and train against the ground truth output $y$, using the mean absolute error (MAE) loss, where we masked out NaN values on the ground truth $y$ during loss computation.

| Model | Params | Layers | Dim. |
|---|---|---|---|
| Reverso-Nano | 200K | 2 | 32 |
| Reverso-Small | 550K | 4 | 64 |
| Reverso | 2.6M | 8 | 128 |

*Table 1.* Architecture configurations.

## 3. Empirical Study

### 3.1. Experimental Setup

We pretrain three versions of Reverso with 200K, 550K and 2.6M parameters. See Table 1 for the model configurations. We train with AdamW (Loshchilov and Hutter, 2019) with maximum learning rate $5 \times 10^{-4}$ using a WSD scheduler (Wen et al., 2025), $\beta_1 = 0.9, \beta_2 = 0.999, \epsilon = 1 \times 10^{-8}$ and weight decay of $0.1$ and we roughly sample 1 million time points per training step with a batch size of 512. Our models take $\{10, 20, 40\}$ H100-hours for a full training run.

**Baselines.** Our baselines include state-of-the-art TSFMs across varying architectures and sizes: Chronos and Chronos-2 (Ansari et al., 2024; 2025), TimesFM-2 and TimesFM-2.5 (Das et al., 2024), PatchTST-FM-r1 (Wen et al., 2026), TiRex (Auer et al., 2025b), FlowState (Graf et al., 2025), Xihe (Sun et al., 2025), Kairos (Feng et al., 2026), Moirai and Moirai-2 (Woo et al., 2024; Liu et al., 2025a), Sundial (Liu et al., 2025c), Toto (Cohen et al., 2025), YingLong (Wang et al., 2025) and Tiny-time Mixers (Ekambaram et al., 2024). Baseline sizes are given in Figure 2 and Table 11.

### 3.2. Main Results

**Zero-shot forecasting.** The Gift-Eval benchmark (Aksu et al., 2024) contains 23 different datasets with 97 different forecasting tasks. As shown in Figure 2 (top), Reverso achieves a competitive MASE value of 0.706 at a modest model size of 2.6M parameters. In particular, we outperform similarly small TSFMs such as Super-Linear (2.5M), FlowState (2.6M) and Tiny-Time Mixers (1M). We also observe

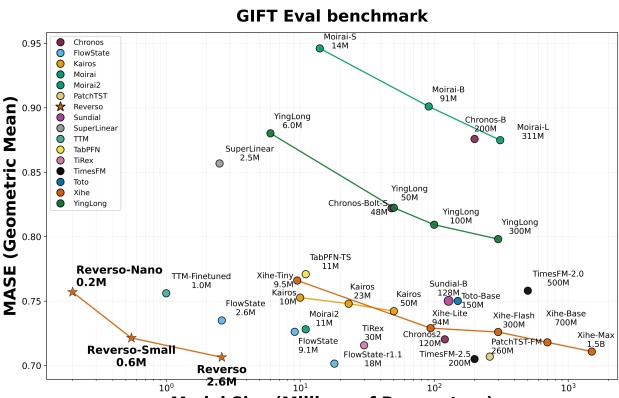

*Figure 2.* MASE on Gift-Eval benchmark

| Model | Params | MASE | | | |
|---|---|---|---|---|---|
| | | **Short** | **Medium** | **Long** | **Average** |
| Xihe-Max | 1.5B | 0.623 | 0.718 | 0.763 | 0.701 |
| TimesFM-2.5 | 200M | 0.626 | 0.724 | 0.751 | 0.700 |
| PatchTST-FM | 260M | **0.616** | 0.722 | 0.745 | 0.694 |
| Flowstate-r1.1 | 18M | 0.633 | 0.720 | **0.736** | 0.696 |
| Reverso | 2.6M | 0.634 | **0.699** | 0.741 | **0.691** |

*Table 2.* Model MASE scores across forecast horizons, averaged across the 21 long sequence datasets in Gift-Eval with all three horizons available.

that our model does particularly well in medium/long horizon forecasting. Table 2 shows the performance of the top TSFMs on the long sequence tasks in Gift-Eval tasks that includes all long horizon forecasting tasks in the benchmark) averaged across all three short/medium/long horizon splits. We see that Reverso achieves strong long sequence point forecasting results, despite being the smallest family of models evaluated on this benchmark. Table 10 of the appendix gives the full results broken down by dataset/domain, while Figure 7 shows some qualitative results on various datasets. Table 11 of the appendix for the full table of numeric results.

We also evaluate on the LTSF (Zeng et al., 2023) test set. On this dataset we outperform Sundial (Liu et al., 2025c), Super-Linear (Nochumsohn et al., 2025), Timer-XL (Liu et al., 2025b) and several other models at a much smaller parameter count, as shown in Figure 5. We report more granular performance numbers in Table 4. These results are strong given that in-domain datasets such as Electricity are part of pretraining datasets of some baselines, such as TiRex and Chronos-2. We explore further results in classification(§ C.1), probabilistic forecasting(§ C.5), anomaly detection(§ C.3), imputation(§ C.4).

**Inference latency.** Reverso is parameter-efficient, but is it also wallclock-efficient for inference? We show that this indeed the case. Figure 3 (left) reports average inference latency and on Gift-Eval, along with peak memory (right) measured for single-sample inference, on a single H100. We find that Reverso variants are faster and more memory

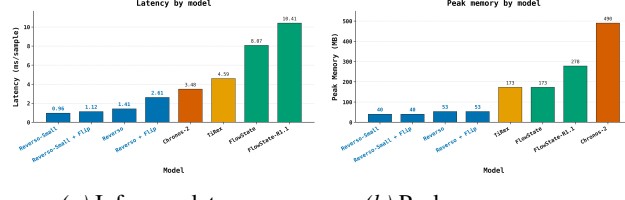

*(a)* Inference latency.  *(b)* Peak memory usage.

*Figure 3.* Inference efficiency on Gift-Eval. Reverso variants achieve lower latency and lower peak memory usage than larger time series foundation models while having the lowest parameters.

efficient than substantially larger TSFMs while remaining competitive in forecasting performance.

## 4. Discussion

The main lesson from Reverso is that competitive zero-shot time-series forecasting does not necessarily require very large TSFMs. Across Gift-Eval and LTSF, Reverso matches or approaches models that are one to three orders of magnitude larger, while also improving inference latency and memory usage. This suggests that, for many workloads represented by current TSFM benchmarks, the limiting factor is not only model scale, but also how effectively model capacity is allocated to the structure of time-series data.

Reverso still has several limitations. First, Reverso is trained primarily as a univariate forecasting model. Chronos-2 has shown that attention can be cleverly utilized to learn cross-channel dependence in multivariate time series. Future work could investigate the potentials of the various sequence mixing layers in multivariate domains. Second, while Reverso's performance on long sequence was near state-of-the-art, its performance on shorter sequences still lagged behind larger TSFMs. Finally, we focus primarily on point prediction, although some applications of interest would benefit from distributional predictions; insofar as conformal methods (Stankeviciute et al., 2021; Sun and Yu, 2025) have also been adopted as a lightweight adaptation to obtain uncertainty bounds for any point time series forecasts, we anticipate such techniques being applicable to obtain uncertainty estimates from Reverso models (see Appendix C.5 for preliminary results in this direction).

## 5. Conclusion

This paper presents Reverso, a family of parameter-efficient time-series foundation models for zero-shot forecasting. Reverso combines multi-scale inputs, hybrid convolutional and linear recurrent sequence mixing, and an attention-based decoder head with a practical training and inference recipe. Across forecasting, classification, and anomaly detection benchmarks, Reverso achieves competitive performance while using substantially fewer parameters and less inference memory than much larger TSFMs. These results suggest that careful architecture and recipe design can significantly shift the TSFM performance–efficiency frontier, making compact foundation models a practical alternative to continued scaling in deployment-constrained settings.

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

# A. Related Works

**Time series foundation models.** Our work is related to the existing research program around time series foundation models (TSFMs), which aim to train domain-general models for time series analysis and forecasting. TimeGPT (Garza et al., 2023), TimesFM (Das et al., 2024), and Lag-LLaMA (Rasul et al., 2023) were some of the first works to show that decoder-only transformers can be utilized to train TSFMs with strong zero-shot forecasting performance. Timer (Liu et al., 2024) and Timer-XL (Liu et al., 2025b) scaled such generative pretraining with dataset size, model size and context length. Moirai (Woo et al., 2024) incorporates a masked encoder to handle multivariate forecasting from various distributions. Chronos (Ansari et al., 2024) fixes the vocabulary of time series patches, while Chronos-2 (Ansari et al., 2025) introduced the group attention mechanism for multivariate forecasting. Xihe (Sun et al., 2025) scales up TSFMs to over a billion parameters with a hierarchical block attention mechanism. PatchTST-FM-r1 (Wen et al., 2026) showed that a generic patched transformer can also achieve competitive results. A complementary line of work reuses large language models directly for time series by reprogramming or aligning them to TS tasks (Zhou et al., 2023; Jin et al., 2023; Chang et al., 2025). However, recent studies suggest that the LLM backbone often provides little benefit over simpler LLM-free baselines (Tan et al., 2024), motivating dedicated TSFMs.

# B. Additional Methodology Details

## B.1. Dataset

**Pretraining dataset.** The time series community has developed a series of commonly-used datasets (Godahewa et al., 2021; Alexandrov et al., 2019; Aksu et al., 2024; Woo et al., 2024), consisting of data from various sources such as weather, traffic, and other domains. We train our models on the GiftEval (Aksu et al., 2024) pretraining dataset, which has become the de facto standard for training TSFMs in recent years. The GiftEval pretraining dataset has around 4.5 million time series with 230 billion time points in total. The dataset however is significantly imbalanced towards datasets such as Buildings900k (Emami et al., 2024), and Era5 (Hersbach et al., 2020). To resolve the imbalance, we precompute the strides on each dataset necessary to achieve a target (roughly uniform) fraction of time series sampled. For each dataset, we target a maximum of $N_{max} = 100000$ samples per epoch, and recompute the strides such that we have at most $N_{max}$ samples from each dataset. Explicitly, for each dataset $\mathcal{D}$ with time series samples $t \in D$ each of length $l_t$, we compute the total sum of lengths as $\sum_{t \in \mathcal{D}} l_t$, and compute the stride for this dataset as $s_{\mathcal{D}} = \left\lceil \frac{\sum_{t \in \mathcal{D}} l_t}{N_{max}} \right\rceil$. We also set an upper limit to 48 samples per time series $t$, to avoid oversampling short datasets. A random start point in each sequence $t$ is chosen at each epoch to ensure sampling across the full pretraining set.

**Data augmentation.** Several techniques for data augmentation have been previously reported to help increase data diversity during pretraining TSFMs. We explored these augmentation techniques and found the following to be useful, which we eventually incorporated into our pretraining recipe: downsampling, amplitude modulation, flip along the $y$ and $x$-axis (i.e., sign inversion and temporal reversal in Moroshan et al. (2025)), censor augmentation and mixup (Ansari et al., 2024), applied in this order. See Figure 4 (left). Downsampling and amplitude modulation are applied at the level of the full sequence. Flip augmentations and censor augmentations are applied on each subsampled sequence of context length $L$ and mixup is applied to the full batch. The full data augmentation pipeline is given in Algorithm 4 of the appendix.

**Synthetic data.** We use synthetic data similar to established baselines (Auer et al., 2025b; Ansari et al., 2024), using methods such as KernelSynth (Ansari et al., 2024), which use Gaussian processes to generate synthetic data. In particular, we define a kernel bank $\mathcal{K}$ (see Table 9 of the appendix), and sample $j \sim U\{1, 5\}$ kernels from $\mathcal{K}$ and compose them using random binary additive or multiplicative operations. This forms a composite kernel $\tilde{\kappa}$. We also sample a mean $\mu$ which follows a linear trend with slope $m \sim U[-0.01, 0.01]$ and intercept $c \sim U[-0.1, 0.1]$ with probability $1/2$ and constant otherwise. We then use $\tilde{\kappa}$ and $\mu$ in a Gaussian process to sample the synthetic time series $t_{syn}$. We also include spike processes (Auer et al., 2025b; Moroshan et al., 2025; Feng et al., 2026) and TSI (Bahrpeyma et al., 2021) as used in Chronos-2, as further described in Algorithms 2 and 3 of the appendix. We generate a total of 1 million synthetic time series sequences with the above algorithm. The maximum sequence length is set to 4096. Figure 4 (right).

## B.2. Inference

**Flip equivariance.** Following prior works (Das et al., 2024), we found it helpful to ensure flip equivariance by passing both the original and flipped context to the model, and then averaging the results, i.e., $\hat{y} = \frac{f(x) - f(-x)}{2}$. While this requires two forward passes of the model, we observe that this reduces forecasting error consistently across multiple benchmarks.

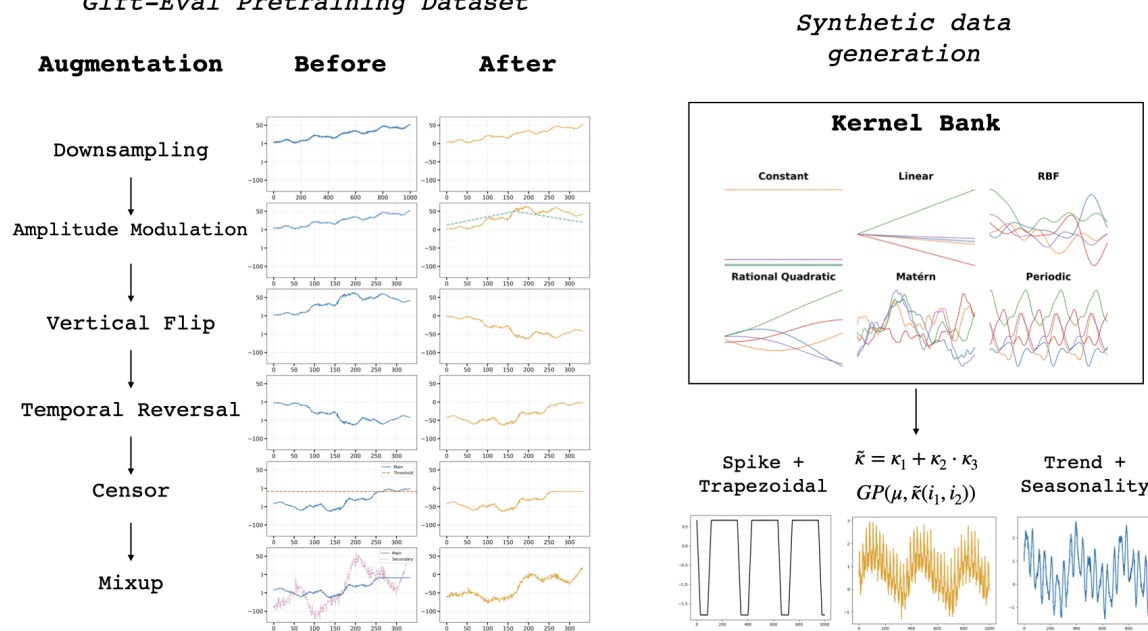

*Figure 4.* Our data augmentation (left) and synthetic data generation (right) pipeline. For data augmentation we apply: downsampling, amplitude modulation, vertical flip, horizontal flip, censor, mixup. For synthetic data generation, we generate data from Gaussian process with randomly selected kernels from a kernel bank, and combine this with spike/trapezoidal patterns as well as processes sampled with trend, seasonality and irregularity.

**Inference-time downsampling.** Given a pretrained TSFM with fixed context length $L$, we generally want to ensure that patterns we wish to capture in the time series have seasonality period $P < L$. The case of $P > L$ potentially results in insufficient information for an effective forecast. Works such as Flowstate (Graf et al., 2025) determine this downsampling factor by rescaling the series with a ratio between the seasonality of the data and a base seasonality of the model. However, such an approach relies heavily on the metadata of the input that might not be available at inference time. We instead use a simple algorithm to determine the downsampling factors using FFT as described below. We first compute FFT of the input sequence to obtain the amplitude spectrum $A(f)$. We then identify the peaks in the spectrum. To distinguish the dominant peak from noise, we enforce a set of criteria, as described in Algorithm 5 and Appendix G. Sequences where the seasonality exceeds the context length $L$ of Reverso are downsampled by a factor of $k$ to $t'$ which is passed as input to the model. Given an original forecast horizon $T$, the model now predicts $\lceil T/k \rceil$ timesteps, which are then upsampled to $T$ by linear interpolation. See Figure 8 of the appendix. Note that this inference-time downsampling strategy is on top of the downsampling we perform for obtaining the multi-scale inputs.

## C. Additional Benchmark Results

We include additional details about our downstream benchmarks.

### C.1. Classification Benchmarks

We evaluate the quality of Reverso representations as features for time-series classification on the UCR(Dau et al., 2019), a univariate archive with 128 datasets, and UEA(Bagnall et al., 2018), a multivariate archive. Following the protocol of (Auer et al., 2025a), the encoder is kept frozen and used purely as a feature extractor: hidden states are collected from every convolution/attention block, max-pooled along the time axis, and concatenated across layers to form a per-series embedding.

We further augment each embedding with two cheap, non-learned signals shown to be useful in (Auer et al., 2025a): patch-wise sample statistics, including mean, standard deviation, minimum, and maximum computed over $k = 256$ non-overlapping patches, and the encoder embedding of the first-order differenced series.

For multivariate UEA datasets, each channel is fed independently through the univariate encoder, and the resulting per-channel embeddings are concatenated. Series shorter than the model context length are upsampled by integer-factor linear interpolation. The final feature vector is then passed to a Random Forest with 300 trees, using `sklearn` defaults otherwise.

Table 3 reports test accuracy on both archives; Reverso matches the best-performing baselines on UCR and UEA without any task-specific fine-tuning. Following (Auer et al., 2025a), we filter out the following datasets in Table 3 (because the authors of (Auer et al., 2025a) encountered processing issues with these datasets): MotorImagery, HandOutlines, StandWalkJump, EigenWorms, and Rock, InsectWingbeat and PLAID. The only difference in the table is TIC which reports the accuracy on the full set of 128 UCR tasks. We did not encounter any processing problems and ran for the full UCR archive in which Reverso also obtained an accuracy of 81%.

*Table 3.* Classification accuracy on UCR and multivariate UEA benchmarks. Higher is better.

| Model | Size | UCR | Multivariate UEA |
|---|---|---|---|
| Reverso | 2.6M | **81%** | **74%** |
| TiRex | 30M | **81%** | **74%** |
| Chronos-Bolt | 205M | 79% | **74%** |
| TIC | – | 80% | – |
| Moirai-Large | 311M | 80% | 70% |
| TimesFM-2.0 | 500M | 79% | 70% |
| Chronos | 200M | 76% | 70% |
| TimesFM-1.0 | 200M | 75% | – |

## C.2. LTSF/TSLib

*Table 4.* Zero-shot forecasting performance (MAE) on LTSF datasets: ETTm1, ETTm2, ETTh1, ETTh2, Electricity and Weather, comparing between Reverso variants against state-of-the-art foundation models. Results represent the averaged MAE across prediction lengths $\{96, 192, 336, 720\}$. Best results are in **bold**, and second-best are underlined. A full set of results are shown in Table 13

| Model | Reverso | Reverso-Small | Reverso-Nano | Sundial-L | Super-Linear | Timer-XL | TiRex | Chronos-2 |
|---|---|---|---|---|---|---|---|---|
| **Params** | 2.6M | 550K | 200K | 444M | 2.6M | 85M | 30M | 120M |
| ETTm1 | 0.367 | 0.376 | 0.382 | 0.369 | 0.389 | 0.392 | 0.365 | **0.359** |
| ETTm2 | 0.304 | 0.309 | 0.311 | 0.315 | 0.325 | 0.336 | 0.302 | **0.291** |
| ETTh1 | **0.404** | **0.404** | 0.416 | 0.420 | 0.416 | 0.417 | 0.417 | 0.405 |
| ETTh2 | 0.365 | 0.370 | 0.384 | 0.387 | 0.386 | 0.388 | **0.362** | 0.367 |
| Electricity | 0.238 | 0.241 | 0.249 | 0.262 | 0.267 | 0.268 | 0.240 | **0.237** |
| Weather | 0.253 | 0.252 | 0.257 | 0.275 | 0.275 | 0.294 | 0.247 | **0.245** |
| **Avg** | 0.322 | 0.325 | 0.333 | 0.338 | 0.343 | 0.349 | 0.322 | **0.317** |
| **Avg Rank** | 2.50 | 3.83 | 5.17 | 6.33 | 6.17 | 7.83 | 2.83 | **1.67** |

We next explore zero-shot transfer results to the LTSF (Zeng et al., 2023) test set. On this dataset we outperform Sundial (Liu et al., 2025c), Super-Linear (Nochumsohn et al., 2025), Timer-XL (Liu et al., 2025b) and several other models at a much smaller parameter count, as shown in Figure 5. We report more granular performance numbers in Table 4, where we follow Sundial (Liu et al., 2025c) and report the mean MAE achieved across various prediction horizons for the datasets of ETTh1, ETTh2, ETTm1, ETTm2, Electricity and Weather.

These results are especially strong given that some of the baselines are quite advantaged compared to Reverso. For example, in-domain datasets such as Electricity enter into the pretraining datasets of TiRex and Chronos-2. Moreover, for models which do not report results on the full benchmark, we impute their scores with the best existing model on each missing dataset.[1] Despite the advantage given to all other models, we observe that Reverso is still the one of the best performing class of models on LTSF.

---

[1]For instance, the values for Electricity for YingLong were imputed using the MAE values obtained by Chronos-2.

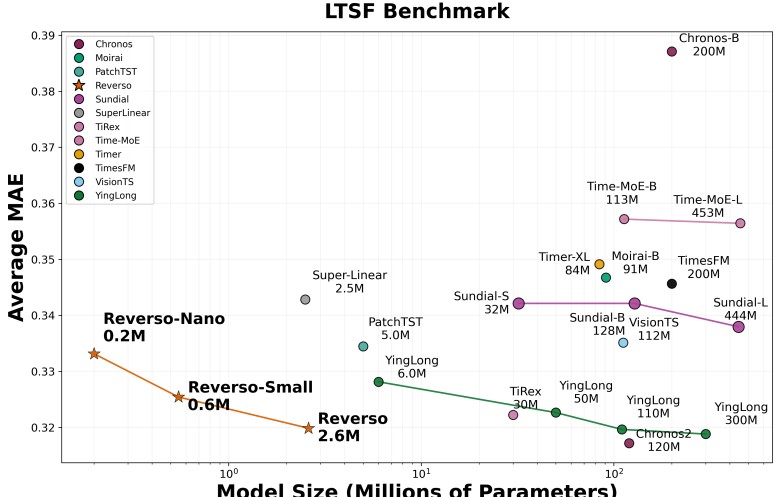

*Figure 5.* LTSF performance vs. Parameter Count. MAE is averaged over the horizons of $\{96, 192, 336, 720\}$ for the datasets ETTh1, ETTh2, ETTm1, ETTm2, Electricity and Weather. For models which are not evaluated on all the datasets (e.g. YingLong did not report results for Electricity), we impute with the other best existing model on that dataset.

## C.3. Anomaly Detection

Table 5 reports zero-shot anomaly detection performance on the UCR Anomaly Archive. We follow the predictive framework of (Liu et al., 2024): rather than training a dedicated detector, we slide the pretrained Reverso forecaster across each series, use each context window to predict the next output segment, and take the forecast error, measured as mean absolute error between prediction and ground truth, as the anomaly score for that segment.

Segments are then ranked by score, and a threshold quantile $q$ is applied so that the top-$q$ fraction of segments is flagged as anomalous. A series is considered detected at level $q$ if the segment overlapping the labeled anomaly falls within this top-$q$ fraction.

We report the number of detected series at $q \in \{1\%, 3\%, 10\%\}$. At every threshold, Reverso detects strictly more anomalies than the baselines, including the tightest $q = 1\%$ regime, where most baseline forecasters miss the labeled anomaly entirely.

*Table 5.* Zero-shot anomaly detection on UCR. Higher detected counts are better. Percentages in parentheses indicate the fraction of UCR anomalies detected by Reverso.

| Threshold | Reverso | Timer | Anomaly Transformer | TimesNet |
|---|---|---|---|---|
| Detected@1% | **88 (35%)** | 51 | 51 | 31 |
| Detected@3% | **126 (50%)** | 110 | 98 | 61 |
| Detected@10% | **188 (75%)** | 172 | 129 | 109 |

## C.4. Imputation

We adopt the imputation protocol of Timer (Liu et al., 2024): each input window of length 192 is divided into $N = 8$ non-overlapping segments of length 24, the first segment is always kept observed, and a fraction $r = 0.5$ of the remaining segments is randomly masked.

Reverso, originally pre-trained as a univariate forecaster, is adapted in a BERT-style masked-reconstruction setting: each channel of a multivariate window is processed independently, instance-normalized on the un-corrupted input, and embedded with the pretrained patch embedding. Embeddings at masked positions are then replaced by a learnable `[MASK]` token before being passed through the encoder. A linear head then each output token back to the input space, and the network is fine-tuned end-to-end with an MSE loss computed only on masked positions. Following Timer, datasets are standardized using train-split statistics and split chronologically: fixed splits for ETT, and $70/10/20$ splits for Weather, ECL, and Traffic. Table 6 reports the resulting test MSE on the seven standard benchmarks; Reverso achieves the lowest MSE on every dataset.

*Table 6.* Imputation MSE on standard time-series benchmarks. Lower is better.

| Model | ETTh1 | ETTh2 | ETTm1 | ETTm2 | Weather | Traffic | ECL |
|-------|-------|-------|-------|-------|---------|---------|-----|
| Reverso | **0.285** | **0.147** | **0.149** | **0.084** | **0.081** | **0.433** | **0.151** |
| Timer | 0.312 | 0.209 | 0.413 | 0.158 | 0.158 | 0.477 | 0.152 |
| TimesNet | 0.289 | 0.253 | 0.373 | 0.112 | 0.127 | 0.624 | 0.209 |

### C.5. Probabilistic Forecasting

To evaluate whether the Reverso backbone can support probabilistic forecasting, we repurpose a pretrained Reverso model by training the decoder head with a quantile regression objective. Table 7 reports CRPS on Gift-Eval splits. Although this experiment does not train the full model from scratch for probabilistic forecasting, Reverso remains competitive with existing TSFMs and obtains the best score on the medium-horizon split.

*Table 7.* Probabilistic forecasting performance measured by CRPS. Lower is better. Reverso uses a pretrained backbone with a quantile-regression decoder head.

| Split | TiRex 30M | FlowState 9.1M | Moirai-2 11M | Kairos 50M | TempoPFN 30M | Reverso 3M |
|-------|-----------|----------------|--------------|------------|--------------|------------|
| Full | **0.489** | 0.502 | 0.516 | 0.548 | 0.533 | 0.499 |
| Short | **0.502** | 0.519 | 0.516 | 0.550 | 0.547 | 0.526 |
| Medium | 0.474 | 0.485 | 0.519 | 0.543 | 0.543 | **0.459** |
| Long | **0.467** | 0.477 | 0.513 | 0.549 | 0.539 | 0.470 |

### C.6. Robustness Across Seeds

We additionally evaluate robustness across random seeds. Table 8 shows that Reverso's final Gift-Eval MASE is stable across four completed runs, with a mean MASE of $0.7115$ and standard deviation of $0.0013$.

*Table 8.* Gift-Eval MASE across random seeds. Lower is better.

| Run Setting | Seed 1 | Seed 2 | Seed 3 | Seed 4 | Mean $\pm$ Std. |
|-------------|--------|--------|--------|--------|-----------------|
| 20% training run | 0.720 | 0.722 | 0.719 | 0.723 | $0.7210 \pm 0.0018$ |
| Full training run | 0.711 | 0.712 | 0.713 | 0.710 | $0.7115 \pm 0.0013$ |

## D. Data generation and augmentation details

### D.1. Synthetic data composition

We make use of standard synthetic data generation practices that has been developed in the community.

KernelSynth(Ansari et al., 2024) introduced the use of Gaussian Process(GP) for synthetic data generation. In particular, we define a kernel bank $\mathcal{K}$, and sample $j \sim U\{1, 5\}$ kernels from $\mathcal{K}$ and compose them using random binary additive or multiplicative operations. This forms a composite kernel $\tilde{\kappa}$. We also sample $\mu$ which follows a linear trend with slope $m \sim U[-0.01, 0.01]$ and intercept $c \sim U[-0.1, 0.1]$ with probability $1/2$ and constant otherwise. We then use $\tilde{\kappa}$ and $\mu$ in a Gaussian process to sample the synthetic time series $t_{syn}$ according to

$$t_{syn} \sim \text{GaussianProcess}(\mu, \tilde{\kappa}(i_1, i_2)).$$

We use the following sets of kernels in our kernel bank $\mathcal{K}$ as shown in Table 9, applied to points $L_{syn}$ evenly spaced points $x, x' \in [0, 1]$. To enable efficient sampling, we use batched Cholesky decomposition. The constant, linear, RBF and Rational Quadratic kernels were introduced in KernelSynth (Ansari et al., 2024). The Matern kernel was used in TempoPFN(Moroshan et al., 2025) as a more robust and accurate representation of how GPs can model real world data. We use the following set of periods $\mathcal{P} = \{24, 48, 96, 168, 336, 672, 7, 14, 30, 60, 365, 730, 4, 26, 52, 6, 12, 40, 10\}$(normalized by time series length $L_{syn}$) to capture patterns of various timescales.

Beyond GP data, we also include spike processes (Auer et al., 2025b; Moroshan et al., 2025; Feng et al., 2026) and TSI (Bahrpeyma et al., 2021) as used in Chronos-2 to help in learning simple trends and periodic patterns.

---

**Algorithm 1** KernelSynth Data Generation

---

**Require:** Length $L$, Kernel bank $\mathcal{K}$ (e.g., RBF, Periodic, Linear, Rational Quadratic), Max kernels $J_{max} = 5$.

1: **Compose Kernel** $\tilde{\kappa}$**:**
2: Sample number of kernels $N \sim \text{Uniform}\{1, J_{max}\}$
3: Sample base kernel $\tilde{\kappa} \sim \mathcal{K}$
4: **for** $i = 2$ to $N$ **do**
5:    Sample next kernel $k' \sim \mathcal{K}$
6:    Sample operation $\oplus \sim \{\text{Add}, \text{Multiply}\}$
7:    $\tilde{\kappa} \leftarrow \tilde{\kappa} \oplus k'$
8: **end for**
9: **Define Mean Function** $\mu(t)$**:**
10: **if** $u \sim U(0, 1) < 0.5$ **then**
11:    *Linear trend:* Sample $m \sim U[-0.01, 0.01]$, $c \sim U[-0.1, 0.1]$
12:    $\mu(t) = m \cdot t + c$
13: **else**
14:    *Constant:* $\mu(t) = 0$ (or sample constant $c$)
15: **end if**
16: **Sample Time Series from Gaussian Process:**
17: Compute covariance matrix $\Sigma \in \mathbb{R}^{L \times L}$ where $\Sigma_{uv} = \tilde{\kappa}(u, v)$
18: Compute mean vector $\mathbf{m} \in \mathbb{R}^L$ where $\mathbf{m}_t = \mu(t)$
19: Sample $t_{syn} \sim \mathcal{N}(\mathbf{m}, \Sigma)$ {Multivariate Gaussian}
20: **return** $t_{syn}$

---

**Algorithm 2** Spike Process Generation, adapted from Kairos (Feng et al., 2026)

---

**Require:** Length $L$, Pattern types $\mathcal{T} = \{$"inverted_u", "spikes"$\}$, Ranges for baseline $[b_{min}, b_{max}]$, period $[p_{min}, p_{max}]$, amplitude $[a_{min}, a_{max}]$, width $[w_{min}, w_{max}]$, and noise $[\sigma_{min}, \sigma_{max}]$.

1: **Sample parameters:**
2: $type \sim \text{Uniform}(\mathcal{T})$, $b \sim \text{Uniform}(b_{min}, b_{max})$, $p \sim \text{Uniform}\{p_{min}, p_{max}\}$
3: $a \sim \text{Uniform}(a_{min}, a_{max})$, $w \sim \text{Uniform}\{w_{min}, w_{max}\}$, $\sigma_\epsilon \sim \text{Uniform}(\sigma_{min}, \sigma_{max})$
4: **Construct trapezoid shape** $e$ **of length** $w$**:**
5: Define $u = \lfloor w/4 \rfloor$, $f = \lfloor w/2 \rfloor$, $d = w - u - f$
6: $e_{up} = \text{linspace}(0, a, u)$, $e_{flat} = \text{constant}(a, f)$, $e_{down} = \text{linspace}(a, 0, d)$
7: $e = [e_{up}; e_{flat}; e_{down}]$
8: **Initialize series:** $x_t = b$ for $t = 1, \ldots, L$
9: $s = -1$ if $type = $ "inverted_u" else $1$
10: **Add periodic patterns:**
11: **for** $i = 0, p, 2p, \cdots < L$ **do**
12:    $len = \min(w, L - i)$
13:    $x_{i:i+len} \leftarrow x_{i:i+len} + s \cdot e_{1:len}$
14: **end for**
15: **Add white noise:** $x \leftarrow x + \epsilon$, where $\epsilon \sim \mathcal{N}(0, \sigma_\epsilon^2)$
16: **return** $x$

---

---

**Algorithm 3** TSI (Trend, Seasonality, Irregularity) Generation, following Chronos-2(Ansari et al., 2025)

---

**Require:** Length $L$, Component probabilities $P_{trend}, P_{seas}, P_{noise}, P_{out}, P_{shift}$, Trend types $\mathcal{T}$, Seasonality periods $\mathcal{P}$, Wave shapes $\mathcal{W}$, Noise distributions $\mathcal{D}$.

1: **Initialize:** $x_t \leftarrow 0$ for $t = 1, \ldots, L$
2: **Add Trend:**
3: **if** $u \sim U(0,1) < P_{trend}$ **then**
4:  Sample trend type $\tau \in \mathcal{T}$ (e.g., linear, exp, poly, piecewise)
5:  Sample parameters $\theta_\tau$ (slope, intercept, degree, etc.)
6:  $x \leftarrow x + f_\tau(t; \theta_\tau)$
7: **end if**
8: **Add Seasonality:**
9: **if** $u \sim U(0,1) < P_{seas}$ **then**
10:  Sample number of components $K \sim U\{1,3\}$
11:  Sample distinct periods $\{p_1, \ldots, p_K\} \subset \mathcal{P}$
12:  **for** $k = 1$ to $K$ **do**
13:   Sample wave form $w \in \mathcal{W}$ (e.g., sine, sawtooth, square)
14:   Sample amplitude $A$ and phase $\phi$
15:   $x_t \leftarrow x_t + A \cdot w\left(\frac{2\pi}{p_k}t + \phi\right)$
16:  **end for**
17: **end if**
18: **Add Irregularity (Noise):**
19: **if** $u \sim U(0,1) < P_{noise}$ **then**
20:  Sample distribution $\mathcal{N} \in \mathcal{D}$ and scale $\sigma$
21:  $x \leftarrow x + \epsilon$, where $\epsilon \sim \mathcal{N}(0, \sigma)$
22: **end if**
23: **Add Anomalies:**
24: **if** $u \sim U(0,1) < P_{out}$ **then**
25:  Add random sparse outliers to $x$
26: **end if**
27: **if** $u \sim U(0,1) < P_{shift}$ **then**
28:  Add random level shifts (step functions) to $x$
29: **end if**
30: **return** $x$

---

*Table 9.* Kernel Bank $\mathcal{K}$ used for Synthetic Data Generation

| Kernel | Formula $\kappa(x, x')$ | Hyperparameters |
|---|---|---|
| Constant | $C$ | $C = 1$ |
| Linear | $\sigma^2 + x \cdot x'$ | $\sigma \in \{0, 1, 10\}$ |
| RBF | $\exp\left(-\frac{\|x-x'\|^2}{2l^2}\right)$ | $l \in \{0.1, 1, 10\}$ |
| Rational Quadratic | $\left(1 + \frac{\|x-x'\|^2}{2\alpha}\right)^{-\alpha}$ | $\alpha \in \{0.1, 1, 10\}$ |
| Matérn | $\frac{2^{1-\nu}}{\Gamma(\nu)}\left(\sqrt{2\nu}\frac{\|x-x'\|}{l}\right)^{\nu} K_{\nu}\left(\sqrt{2\nu}\frac{\|x-x'\|}{l}\right)$ | $\nu \in \{0.5, 1.5, 2.5\}, l \in \{0.1, 1, 10\}$ |
| Periodic | $\exp\left(-2\sin^2(\pi\|x-x'\|/p)\right)$ | $p \in \mathcal{P}$ |

## D.2. Data augmentation specifics

Here we present the various data augmentation strategies that we found to have been helpful in improving the data diversity during training. We demonstrate this in a full pipeline detailed in Algorithm 4. Our pipeline applies transformations at both the instance level (sequentially) and the batch level (Mixup).

1. **Downsampling:** To allow the model to learn features across varying temporal resolutions, we downsample the raw time series $t$ by a factor $k$. This effectively compresses long-term dependencies into the context window $L$.

2. **Amplitude Modulation:** We multiply $t$ by a piecewise linear function. We follow the implementation from TiRex but sample just a single intermediate changepoint.

3. **Flips:** We apply random sign flips (inverting the y-axis) and temporal reversals (flipping the x-axis). This follows the implementation of TempoPFN (Moroshan et al., 2025).

4. **Censoring:** The series is clipped from both the top and the bottom. This effectively applies a per-sample thresholding which reduces the effect of anomalies on training.

5. **Batch Mixup:** We apply Mixup (Ansari et al., 2024) at the batch level, creating a convex interpolation between samples.

The formal procedure for generating a single training batch is detailed in Algorithm 4.

---

**Algorithm 4** Pretraining Data Augmentation Pipeline

---

**Input:** Dataset $\mathcal{D}$, Batch size $B$, Context length $L$

**Hyperparameters:**
  $p(\texttt{downsample}), [k_{min}, k_{max}]$ (Downsample probability, range of downsample ratios)
  $p(\texttt{modulate}), p(\texttt{flip-x}), p(\texttt{flip-y})$ (Amp. Mod., Flip-$x$, Flip-$y$ probabilities)
  $p(\texttt{censor})$ (Censor prob), $\alpha$ (Mixup beta param)

Initialize batch $\mathcal{B} \leftarrow \emptyset$
**while** $|\mathcal{B}| < B$ **do**
  Sample raw time series $X$ from $\mathcal{D}$
  {1. Multi-scale Downsampling}
  **if** sample $u \sim U(0,1) < p_d$ **then**
    Sample stride $k \sim U_{\text{int}}(k_{min}, k_{max})$
    $X \leftarrow \text{Downsample}(X, k)$
  **end if**
  {2. Amplitude Modulation}
  **if** sample $u \sim U(0,1) < p(\texttt{modulate})$ **then**
    Sample changepoint $x_2 \subset \{1, \ldots, \texttt{len}(X) - 2\}$. Set $x_1 = 0, x_3 = \texttt{len}(X) - 1$
    Sample scalar $\{y_1, y_2, y_3\} \sim \mathcal{N}(1, 0.5)$
    Piecewise linear $f(x)$ connecting $(x_1, y_1), (x_2, y_2), (x_3, y_3)$
    $X \leftarrow X \cdot f(x)$
  **end if**
  {3. Slicing to context length}
  $T_{len} \leftarrow \texttt{len}(X)$
  **if** $T_{len} > L$ **then**
    Sample $t_{start} \sim U_{\text{int}}(0, T_{len} - L)$
    $x_{seq} \leftarrow X[t_{start} : t_{start} + L]$ (on the next iteration we start at $t_{start} + L$).
  **else**
    $x_{seq} \leftarrow \text{Pad}(X, L)$
  **end if**
  {4. Flip Augmentations}
  **if** sample $u \sim U(0,1) < p(\texttt{flip-y})$ **then**
    $x_{seq} \leftarrow -x_{seq}$ {Sign Inversion}
  **end if**
  **if** sample $u \sim U(0,1) < p(\texttt{flip-x})$ **then**
    $x_{seq}[i] \leftarrow x_{seq}[L - i - 1]$ {Temporal Reversal}
  **end if**
  {5. Censor Augmentation}
  **if** sample $u \sim U(0,1) < p(\texttt{censor})$ **then**
    Sample $q \sim U(0,1)$
    Compute threshold $c \leftarrow \text{Quantile}(x_{seq}, q)$
    $\texttt{direction} \sim \{\texttt{top, bottom, none}\}$
    **if** $\texttt{direction} = \texttt{top}$ **then**
      $x_{seq} \leftarrow \min(x_{seq}, c)$
    **else if** $\texttt{direction} = \texttt{bottom}$ **then**
      $x_{seq} \leftarrow \max(x_{seq}, c)$
    **end if**
  **end if**
  Add $x_{seq}$ to $\mathcal{B}$
**end while**
{6.Mixup}
Construct permutation $\pi$ of $\{0, \ldots, B-1\}$, $\tilde{\mathcal{B}} = \mathcal{B}[\pi]$
Sample $\lambda \sim \text{Beta}(\alpha, \alpha)$
**for** $i = 1$ **to** $B$ **do**
  Sample $\lambda \sim \text{Beta}(\alpha, \alpha)$
  $\mathcal{B}[i] \leftarrow \lambda \mathcal{B}[i] + (1 - \lambda)\tilde{\mathcal{B}}[i]$
**end for**
**return** $\mathcal{B}$

---

# E. Extended results on Gift-Eval

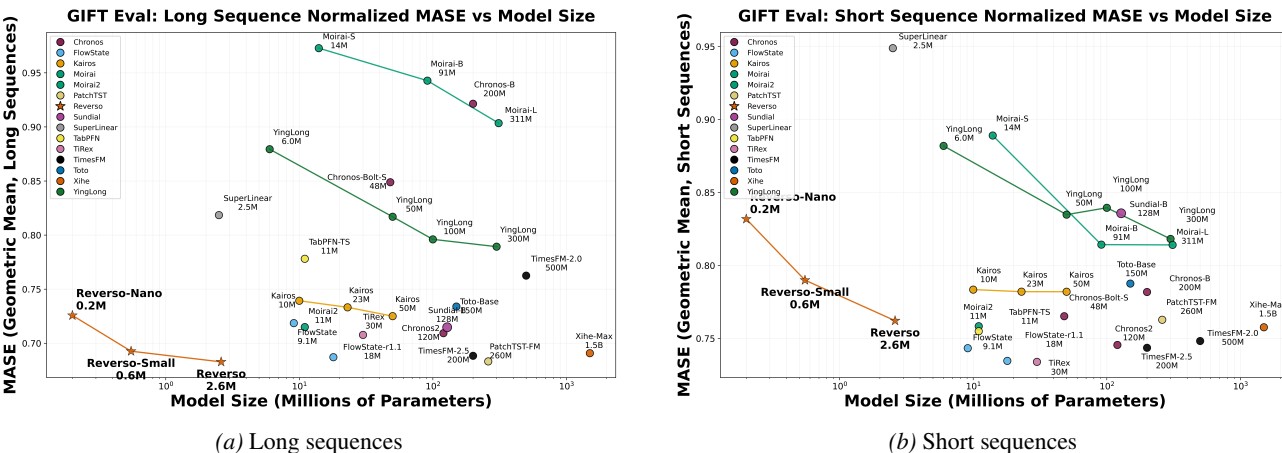

*(a)* Long sequences
*(b)* Short sequences

*Figure 6.* Zero-shot performance on the Gift-Eval benchmark for (a) long sequences (average length at least 2048) and (b) short sequences.

*Table 10.* Full results on Gift-Eval achieved by Reverso, with an overall MASE of 0.706.

| Dataset | Domain | Var | MASE | Dataset | Domain | Var | MASE |
|---|---|---|---|---|---|---|---|
| loop_seattle/5T/short | Transport | 1 | 0.5554 | ett1/H/short | Energy | 7 | 0.8013 |
| loop_seattle/5T/medium | Transport | 1 | 0.7422 | ett1/H/medium | Energy | 7 | 1.2408 |
| loop_seattle/5T/long | Transport | 1 | 0.7978 | ett1/H/long | Energy | 7 | 1.3574 |
| loop_seattle/D/short | Transport | 1 | 0.8752 | ett1/W/short | Energy | 7 | 1.4496 |
| loop_seattle/H/short | Transport | 1 | 0.8099 | ett2/15T/short | Energy | 7 | 0.7483 |
| loop_seattle/H/medium | Transport | 1 | 0.8988 | ett2/15T/medium | Energy | 7 | 0.8739 |
| loop_seattle/H/long | Transport | 1 | 0.8683 | ett2/15T/long | Energy | 7 | 0.9145 |
| m_dense/D/short | Transport | 1 | 0.6762 | ett2/D/short | Energy | 7 | 1.3782 |
| m_dense/H/short | Transport | 1 | 0.8112 | ett2/H/short | Energy | 7 | 0.7069 |
| m_dense/H/medium | Transport | 1 | 0.6777 | ett2/H/medium | Energy | 7 | 0.9491 |
| m_dense/H/long | Transport | 1 | 0.6868 | ett2/H/long | Energy | 7 | 0.9810 |
| sz_taxi/15T/short | Transport | 1 | 0.5437 | ett2/W/short | Energy | 7 | 0.7283 |
| sz_taxi/15T/medium | Transport | 1 | 0.5398 | hierarchical_sales/D/short | Sales | 1 | 0.7474 |
| sz_taxi/15T/long | Transport | 1 | 0.5133 | hierarchical_sales/W/short | Sales | 1 | 0.7429 |
| sz_taxi/H/short | Transport | 1 | 0.5713 | hospital/M/short | Healthcare | 1 | 0.7872 |
| bitbrains_fast_storage/5T/short | Web/CloudOps | 2 | 0.6672 | jena_weather/10T/short | Nature | 21 | 0.2800 |
| bitbrains_fast_storage/5T/medium | Web/CloudOps | 2 | 0.9861 | jena_weather/10T/medium | Nature | 21 | 0.6141 |
| bitbrains_fast_storage/5T/long | Web/CloudOps | 2 | 0.8726 | jena_weather/10T/long | Nature | 21 | 0.6448 |
| bitbrains_fast_storage/H/short | Web/CloudOps | 2 | 0.9920 | jena_weather/D/short | Nature | 21 | 1.3206 |
| bitbrains_rnd/5T/short | Web/CloudOps | 2 | 1.6267 | jena_weather/H/short | Nature | 21 | 0.5175 |
| bitbrains_rnd/5T/medium | Web/CloudOps | 2 | 4.4659 | jena_weather/H/medium | Nature | 21 | 0.7750 |
| bitbrains_rnd/5T/long | Web/CloudOps | 2 | 3.3323 | jena_weather/H/long | Nature | 21 | 0.9717 |
| bitbrains_rnd/H/short | Web/CloudOps | 2 | 5.8143 | kdd_cup_2018/D/short | Nature | 1 | 1.1942 |
| bizitobs_application/10S/short | Web/CloudOps | 2 | 1.1003 | kdd_cup_2018/H/short | Nature | 1 | 0.9422 |
| bizitobs_application/10S/medium | Web/CloudOps | 2 | 1.6500 | kdd_cup_2018/H/medium | Nature | 1 | 1.0369 |
| bizitobs_application/10S/long | Web/CloudOps | 2 | 3.2409 | kdd_cup_2018/H/long | Nature | 1 | 1.0290 |
| bizitobs_l2c/5T/short | Web/CloudOps | 7 | 0.2819 | m4_daily/D/short | Econ/Fin | 1 | 3.3635 |
| bizitobs_l2c/5T/medium | Web/CloudOps | 7 | 0.4443 | m4_hourly/H/short | Econ/Fin | 1 | 0.8070 |
| bizitobs_l2c/5T/long | Web/CloudOps | 7 | 0.4675 | m4_monthly/M/short | Econ/Fin | 1 | 0.9282 |
| bizitobs_l2c/H/short | Web/CloudOps | 7 | 0.4603 | m4_quarterly/Q/short | Econ/Fin | 1 | 1.2281 |
| bizitobs_l2c/H/medium | Web/CloudOps | 7 | 0.4884 | m4_weekly/W/short | Econ/Fin | 1 | 2.0325 |
| bizitobs_l2c/H/long | Web/CloudOps | 7 | 0.5518 | m4_yearly/A/short | Econ/Fin | 1 | 3.2512 |
| bizitobs_service/10S/short | Web/CloudOps | 2 | 0.7742 | restaurant/D/short | Sales | 1 | 0.6893 |
| bizitobs_service/10S/medium | Web/CloudOps | 2 | 0.9437 | saugeen/D/short | Nature | 1 | 2.7549 |
| bizitobs_service/10S/long | Web/CloudOps | 2 | 1.3505 | saugeen/M/short | Nature | 1 | 0.7482 |
| car_parts/M/short | Sales | 1 | 0.8622 | saugeen/W/short | Nature | 1 | 1.2917 |
| covid_deaths/D/short | Healthcare | 1 | 33.8087 | solar/10T/short | Energy | 1 | 1.1933 |
| electricity/15T/short | Energy | 1 | 1.0005 | solar/10T/medium | Energy | 1 | 0.8573 |
| electricity/15T/medium | Energy | 1 | 0.8273 | solar/10T/long | Energy | 1 | 0.8991 |
| electricity/15T/long | Energy | 1 | 0.8759 | solar/D/short | Energy | 1 | 1.0483 |
| electricity/D/short | Energy | 1 | 1.4494 | solar/H/short | Energy | 1 | 0.8501 |
| electricity/H/short | Energy | 1 | 0.9798 | solar/H/medium | Energy | 1 | 0.8548 |
| electricity/H/medium | Energy | 1 | 1.0529 | solar/H/long | Energy | 1 | 0.9507 |
| electricity/H/long | Energy | 1 | 1.1888 | solar/W/short | Energy | 1 | 1.3170 |
| electricity/W/short | Energy | 1 | 1.5892 | temperature_rain/D/short | Nature | 1 | 1.3966 |
| ett1/15T/short | Energy | 7 | 0.6784 | us_births/D/short | Healthcare | 1 | 0.3515 |
| ett1/15T/medium | Energy | 7 | 1.0117 | us_births/M/short | Healthcare | 1 | 0.8479 |
| ett1/15T/long | Energy | 7 | 1.0146 | us_births/W/short | Healthcare | 1 | 1.0146 |
| ett1/D/short | Energy | 7 | 1.6182 | | | | |

*Table 11.* MASE scores versus parameter size breakdowns for models compared in the Gift-Eval benchmark in Figure 2. Note that the data leaked version of Chronos-2 has been trained on datasets within Gift-Eval and hence is not included within Figure 2 for fair zero-shot comparison, but is still a strong frame of reference for SOTA foundation models.

| Family | Model | Params | MASE |
|---|---|---|---|
| TimesFM | TimesFM-2.5 | 200M | 0.705 |
|  | TimesFM-2.0 | 500M | 0.758 |
| PatchTST-FM-r1 | PatchTST-FM | 260M | 0.707 |
| Xihe | Xihe-Max | 1.5B | 0.711 |
|  | Xihe-Base | 700M | 0.718 |
|  | Xihe-Flash | 300M | 0.726 |
|  | Xihe-Lite | 94M | 0.729 |
|  | Xihe-Tiny | 9.5M | 0.766 |
| Reverso | Reverso | 2.6M | 0.706 |
|  | Reverso-Small | 0.6M | 0.721 |
|  | Reverso-Nano | 0.2M | 0.757 |
| TiRex | TiRex | 30M | 0.716 |
| Chronos | Chronos2(Data leakage) | 120M | 0.698 |
|  | Chronos2 | 120M | 0.720 |
|  | Chronos-Bolt-S | 48M | 0.822 |
|  | Chronos-B | 200M | 0.876 |
| FlowState | FlowState-9.1M | 9.1M | 0.726 |
|  | FlowState-2.6M | 2.6M | 0.735 |
|  | FlowState-r1.1 | 18M | 0.701 |
| Kairos | Kairos-50M | 50M | 0.742 |
|  | Kairos-23M | 23M | 0.748 |
|  | Kairos-10M | 10M | 0.753 |
| Toto | Toto-Base | 150M | 0.750 |
| Sundial | Sundial-B | 128M | 0.750 |
| TTM | TTM-Finetuned | 1.0M | 0.756 |
| TabPFN | TabPFN-TS | 11M | 0.771 |
| YingLong | YingLong-300M | 300M | 0.798 |
|  | YingLong-110M | 110M | 0.809 |
|  | YingLong-50M | 50M | 0.822 |
|  | YingLong-6M | 6.0M | 0.880 |
| SuperLinear | SuperLinear | 2.5M | 0.857 |
| Moirai | Moirai-L | 311M | 0.875 |
|  | Moirai-B | 91M | 0.901 |
|  | Moirai-S | 14M | 0.946 |
|  | Moirai2 | 11M | 0.728 |

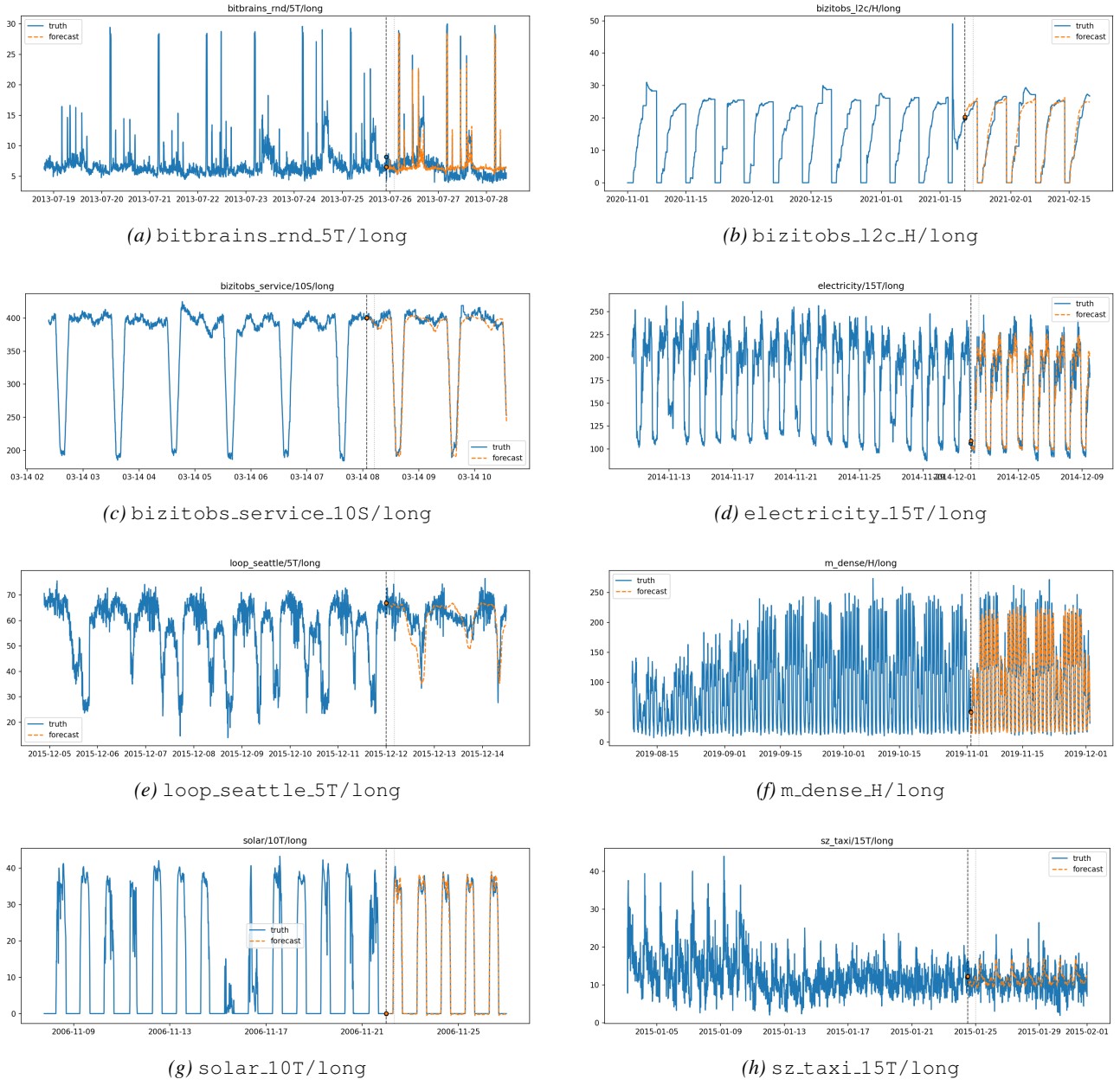

*Figure 7.* Qualitative results visualizing zero-shot forecasts by Reverso on various tasks within Gift-Eval. Reverso is able to perform long horizon forecasting, accurately capturing patterns at multiple frequency scales. The grey vertical dotted line denotes the length of a single autoregressive prediction.

*Table 12.* Datasets used for Table 2, a subset of GiftEval which have all three short/medium/long forecasting horizons.

| Dataset | Frequency |
|---|---|
| bitbrains_fast_storage | 5T |
| bitbrains_rnd | 5T |
| bizitobs_application | 10S |
| bizitobs_l2c | 5T |
| bizitobs_l2c | H |
| bizitobs_service | 10S |
| electricity | 15T |
| electricity | H |
| ett1 | 15T |
| ett1 | H |
| ett2 | 15T |
| ett2 | H |
| jena_weather | 10T |
| jena_weather | H |
| kdd_cup_2018 | H |
| loop_seattle | 5T |
| loop_seattle | H |
| m_dense | H |
| solar | 10T |
| solar | H |
| sz_taxi | 15T |

# F. Detailed results for LTSF/TSLib

*Table 13.* Results across each dataset for the models in Figure 5, averaged across the horizons $\{96, 192, 336, 720\}$. Red values represent missing data, imputed using the best available model across each horizon.

| Family | Model | Params | ETTh1 | ETTh2 | ETTm1 | ETTm2 | Elec. | Weather | Avg |
|---|---|---|---|---|---|---|---|---|---|
| Chronos | Chronos2-120M | 120M | 0.405 | 0.367 | 0.359 | 0.291 | 0.237 | 0.245 | 0.317 |
| | Chronos-B | 200M | 0.468 | 0.410 | 0.500 | 0.350 | 0.279 | 0.315 | 0.387 |
| YingLong | YingLong-300M | 300M | 0.407 | 0.370 | 0.358 | 0.296 | 0.237 | 0.245 | 0.319 |
| | YingLong-110M | 110M | 0.408 | 0.366 | 0.356 | 0.296 | 0.237 | 0.255 | 0.320 |
| | YingLong-50M | 50M | 0.408 | 0.370 | 0.366 | 0.298 | 0.237 | 0.257 | 0.323 |
| | YingLong-6M | 6.0M | 0.412 | 0.382 | 0.368 | 0.302 | 0.237 | 0.268 | 0.328 |
| Reverso | Reverso | 2.6M | 0.404 | 0.365 | 0.367 | 0.304 | 0.238 | 0.253 | 0.322 |
| | Reverso-Small | 0.6M | 0.404 | 0.370 | 0.376 | 0.309 | 0.241 | 0.252 | 0.325 |
| | Reverso-Nano | 0.2M | 0.416 | 0.384 | 0.382 | 0.311 | 0.249 | 0.257 | 0.333 |
| TiRex | TiRex-30M | 30M | 0.417 | 0.362 | 0.365 | 0.302 | 0.240 | 0.247 | 0.322 |
| PatchTST | PatchTST | 5.0M | 0.431 | 0.379 | 0.381 | 0.315 | 0.237 | 0.264 | 0.334 |
| VisionTS | VisionTS | 112M | 0.414 | 0.375 | 0.372 | 0.321 | 0.237 | 0.292 | 0.335 |
| Sundial | Sundial-L | 444M | 0.419 | 0.387 | 0.369 | 0.315 | 0.262 | 0.275 | 0.338 |
| | Sundial-S | 32M | 0.418 | 0.387 | 0.388 | 0.324 | 0.265 | 0.271 | 0.342 |
| | Sundial-B | 128M | 0.434 | 0.387 | 0.377 | 0.320 | 0.265 | 0.270 | 0.342 |
| SuperLinear | Super-Linear | 2.5M | 0.415 | 0.386 | 0.388 | 0.325 | 0.267 | 0.275 | 0.343 |
| TimesFM | TimesFM | 200M | 0.444 | 0.406 | 0.419 | 0.347 | 0.237 | 0.222 | 0.346 |
| Moirai | Moirai-B | 91M | 0.419 | 0.382 | 0.385 | 0.337 | 0.275 | 0.282 | 0.347 |
| Timer | Timer-XL | 84M | 0.417 | 0.388 | 0.392 | 0.336 | 0.268 | 0.294 | 0.349 |
| Time-MoE | Time-MoE-L | 453M | 0.420 | 0.415 | 0.406 | 0.361 | 0.237 | 0.300 | 0.356 |
| | Time-MoE-B | 113M | 0.424 | 0.404 | 0.415 | 0.365 | 0.237 | 0.297 | 0.357 |

## G. Downsampling Algorithm

---

**Algorithm 5** Downsampling

---

**Require:** Time series $x$, Context length $L$.
**Require:** Hyperparameters: Dominance ratio $\alpha$, Significance threshold $\beta$, Min periods in window $M$.
1: Compute amplitude spectrum $A(f) = |\text{FFT}(x)|$
2: Identify peaks: $p_1 \leftarrow \max_{f>0} A(f)$ at frequency $f_1$, $\quad p_2 \leftarrow \max_{f>0, f \neq f_1} A(f)$
3: Compute stats: $p_{DC} \leftarrow A(0)$, $\quad \mu_A \leftarrow \text{mean}(A)$, $\quad \sigma_A \leftarrow \text{std}(A)$
4: **Check Seasonality Significance:**
5: **if** $p_1 \geq \alpha \cdot p_2$ **and** $p_1 \geq p_{DC}$ **and** $p_1 \geq \mu_A + \beta \cdot \sigma_A$ **then**
6: $\quad$ Calculate primary period $S \leftarrow 1/f_1$
7: $\quad$ Compute stride $k \leftarrow \lfloor \frac{M \cdot S}{L} \rfloor$
8: $\quad$ **if** $k > 1$ **then**
9: $\quad\quad$ **return** Downsampled series $x' = [x_0, x_k, x_{2k}, \dots]$
10: $\quad$ **end if**
11: **end if**
12: **return** Original series $x$

---

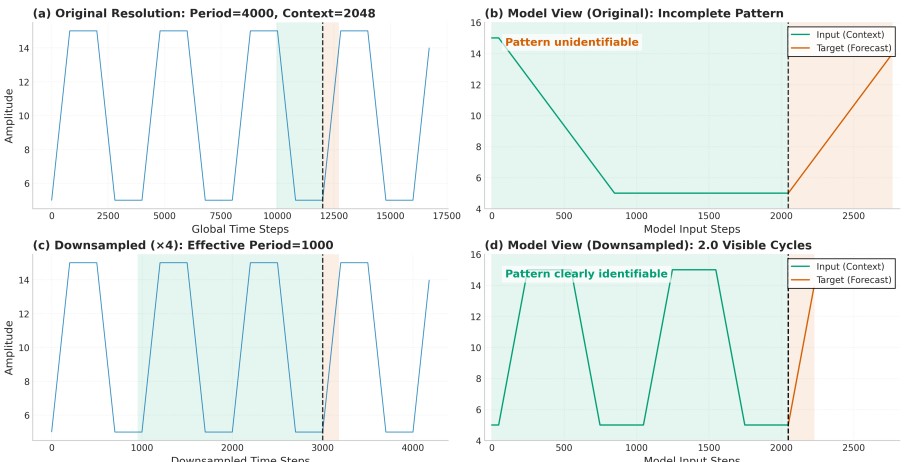

*Figure 8.* Downsampling Comparison. In this example, consider an input with period 4000 to a model with context length 2048, tasked with forecasting the next 720 points. In (b), the model does not have enough information to forecast the rising part of the trapezoid. However, through downsampling the input, multiple full periods now fit into the context window and the model can forecast more accurately.

Let $p_1$ be the amplitude of the highest peak at frequency $f_1$, and $p_2$ be the amplitude of the second highest peak. Let $p_{DC}$ denote the DC component (amplitude at $f = 0$), and let $\mu_A, \sigma_A$ be the mean and standard deviation of the spectral amplitudes, respectively. We consider the seasonality at $f_1$ significant if and only if all the following conditions are met:

$$p_1 \geq \alpha \cdot p_2 \tag{1}$$

$$p_1 \geq p_{DC} \tag{2}$$

$$p_1 \geq \mu_A + \beta \cdot \sigma_A \tag{3}$$

Equation 1 ensures a single dominant frequency exists, mitigating ambiguity from multi-scale seasonality which we leave for future work. Equation 2 ensures the seasonality is stronger than the trend component, and Equation 3 provides statistical confidence that the signal is not merely noise.

If these conditions are satisfied, we calculate the primary period $S = 1/f_1$. To ensure the model captures sufficient temporal

context, we compute a downsampling stride $k$ such that at least $M$ full periods fit within the fixed context window $L$:

$$k = \left\lfloor \frac{MS}{L} \right\rfloor \tag{4}$$

The input sequence is then downsampled by taking every $k$-th point, effectively expanding the receptive field of the model to cover $k \cdot L$ time steps while maintaining the fixed input dimension $L$. If the spectral peaks do not meet the criteria, we do not downsample. We find that typically applying this downsampling algorithm to single time series samples at each time has high variance, resulting in different downsampling ratios for each sequence and in practice we average the downsampling ratios across the same frequency within the same dataset. We use $\alpha = 2, \beta = 4, M = 8$. Downsampling is not applied to short term forecast for which the forecast horizon is significantly shorter than the seasonality since this reduces the resolution of the predictions without capturing further seasonality information.

## H. Ablations

We perform ablations across various architectural, dataset, and inference-strategy choices. Due to compute constraints we perform ablations on a smaller portion of the training set. Note that Reverso is quite robust across seeds (0.002-0.003 difference across seeds for the smaller dataset, see Appendix C.6), and hence any differences greater than this are considered significant.

**Architecture.** The multiscale downsampling of the input sequence before the embedding layer aims to capture information at various frequency scales. This also extends the effective context window of Reverso, capturing long range dependencies. Using $C = 4$ frequency channels, we downsample the input sequence by a factor of up to $8\times$, improving long horizon forecasting MASE on Gift-Eval across the three model sizes of Reverso-Nano(200K), Reverso-Small(550K) and Reverso(2.6M). Table 14 shows the importance of this multi-scale input strategy, which to the best of our knowledge is novel among TSFMs.

| Model | Long MASE | Full MASE |
|---|---|---|
| Reverso-Nano | 0.766 | 0.757 |
| *w/o. multi-scale input* | 0.783 | 0.760 |
| Reverso-Small | 0.742 | 0.722 |
| *w/o. multi-scale input* | 0.754 | 0.726 |
| Reverso | 0.741 | 0.706 |
| *w/o. multi-scale input* | 0.749 | 0.711 |

*Table 14.* Effect of multiscale downsampling on Gift-Eval MASE. Lower is better.

| Sequence Module | Long MASE | Short MASE | Overall MASE |
|---|---|---|---|
| DeltaNet | 0.706 | 0.792 | 0.732 |
| Gated DeltaProduct | 0.711 | 0.793 | 0.735 |
| Gated DeltaNet | 0.708 | 0.782 | 0.730 |
| Long Convolution | 0.708 | 0.799 | 0.735 |
| Gated Linear Attention | 0.726 | 0.817 | 0.753 |
| Attention (`sin-cos`) | 0.729 | 0.840 | 0.762 |
| Attention (`RoPE`) | 0.719 | 0.824 | 0.750 |
| Conv + DeltaNet (Reverso) | 0.700 | 0.786 | 0.725 |
| *w/o. attention decoder* | 0.719 | 0.789 | 0.740 |

*Table 15.* Architectural ablations.

How much does our hybrid sequence mixing of convolutions and DeltaNet help for time series? In Table 15, we report the MASE achieved by different instances of our model using the different sequence mixing layers. Across each model, we keep the number of layers fixed at 8 and sequence mixing dimension at 128. We find that for non-hybrid models, DeltaNet (Schlag et al., 2021) and Gated DeltaNet (Yang et al., 2025) achieves the low loss with few parameter counts compared to layers like Gated Linear Attention (Yang et al., 2024a) and DeltaProduct (Siems et al., 2025). Overall, linear attention and convolution methods consistently outperform full attention. Hybrid models that combine long convolutions with linear RNN layers ultimately perform best. We provide further ablations in layer ordering in Table 18 and mechanistic insights into their learned properties in Appendix I.

Table 15 (bottom) shows ablation studies on our attention decoder head, where we replace the attention mechanism with a simple (bi)linear layer. For the simple linear layer, the hidden states $x^{(n)} \in \mathbb{R}^{L \times d}$ after the last Reverso block are projected to the output with two linear projections $W_1 \in \mathbb{R}^{d \times 1}$ and $W_2 \in \mathbb{R}^{p \times L}$ with the following transformation $\hat{y} = W_2 x^{(n)} W_1$. We observe that the attention mechanism at the decoder boosts overall performance, in particular helps to capture long range dependencies.

| Method | MASE |
|---|---|
| Baseline | 0.738 |
| w/o mixup | 0.740 |
| w/o downsample | 0.740 |
| w/o temp rev | 0.740 |
| w/o flip | 0.739 |
| w/o censor | 0.738 |
| w/o amp mod | 0.737 |
| w/o any data augmentation | 0.755 |
| w/o synthetic data | 0.786 |

*Table 16.* Data pipeline ablations.

**Data augmentation and synthetic data.** Table 16 shows a leave-one-out experiment for our data augmentation/synthetic data strategies, where we train Reverso (on a smaller training set) while removing each one of the components {`mixup, downsample, temporal reversal(flip-x), vertical flip(flip-y), censor, amplitude modulation`}. We find that our training recipe is robust to the setting of individual data augmentation techniques, where removing a single data augmentation does not significantly hurt pre-training. But at the same time, the usage of augmentations remain necessary, and ablating them altogether is detrimental. Synthetic data also shows significant benefit, even when present in small ratios.

| Method | Short Seq | Long Seq | Short Term | Med Term | Long Term | Overall |
|---|---|---|---|---|---|---|
| Baseline | 0.764 | 0.682 | 0.696 | 0.699 | 0.741 | 0.706 |
| w/o downsampling | 0.764 | 0.700 | 0.696 | 0.723 | 0.778 | 0.719 |
| No flip | 0.776 | 0.686 | 0.705 | 0.702 | 0.745 | 0.713 |
| Flip equivariance | 0.764 | 0.682 | 0.696 | 0.699 | 0.741 | 0.706 |

*Table 17.* Inference strategy ablations on Gift-Eval.

**Inference.** Finally, we analyze in Table 17 (top) the different effects of downsampling and flip equivariance methods described in Section B.2 in forecasting. We find that downsampling helps bring long range dependencies into the context window of our model, improving the medium and long term forecast performance.

Flip equivariance helps more on short sequences and short horizon forecasting. In Table 17 (bottom) the effect of applying the flip-equivariance strategy, which generally helps to reduce MASE, with more prominent gains observed in short sequence/horizon forecasting. This strategy acts as an inference-time ensembling method, which improves performance using just two forward passes of the same model while remaining competitive in the frontier of performance efficiency tradeoff (Figure 3).

| Architecture | Layer Sequence | MASE |
|---|---|---|
| Reverso | Conv→DN→Conv→DN→Conv→DN→Conv→DN | 0.720 |
| Variant 1 | DN→Conv→DN→Conv→DN→Conv→DN→Conv | 0.724 |
| Variant 2 | Conv→DN→DN→DN→DN→DN→DN→DN | 0.725 |
| Variant 3 | Conv→Conv→Conv→Conv→DN→DN→DN→DN | 0.730 |

*Table 18.* MASE performance for different architecture variants.

**Hybrid layer ordering** We provide further ablations on the design of our hybrid model in Table 18, trained to an intermediate checkpoint, across different types of layer orderings between the long convolutions and Deltanet layers. We observe empirically that the (`Conv → Deltanet`)$\times n_{layers}$ format performs better. In the next section, we provide a simple mechanistic analysis to understand the learned properties of the convolution and Deltanet layers.

# I. Mechanistic analysis

To understand the learned properties of Reverso, We conducted an analysis of the long convolution filters and DeltaNet states on Reverso-Nano. First, we provide a visualization of the convolution filters in Figure 9.

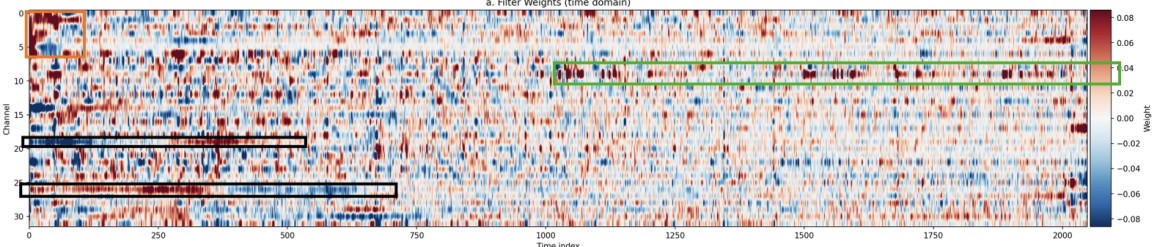

*Figure 9.* Filter visualizations for the first convolution layer of Reverso-Nano.

We observe several interesting phenomena: the orange box indicates a set of weights that are strongly biased towards recent temporal positions. The black boxes indicate low pass filters on frequency scales of around a few hundred time steps.

|                | Conv layer 1 | Conv layer 2 | Conv layer 3 | Conv layer 4 |
|----------------|--------------|--------------|--------------|--------------|
| Min period     | 98           | 293          | 4            | 5            |
| Median period  | 244          | 293          | 204          | 163          |

*Table 19.* Minimum and median periods across convolutional layers.

We also performed FFT on the convolution filters across each Reverso(2.6M) layer (see Table 19) and observe that later convolution layers act as higher frequency filters. We believe that the alternating structure enables the convolutions to help each DeltaNet block operate at varying frequency scales.

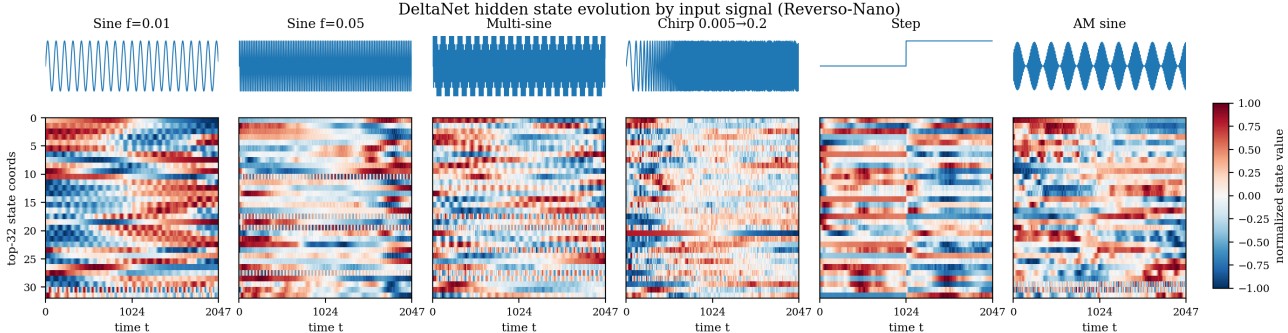

*Figure 10.* State evolution of of the Deltanet states in Reverso-Nano. Across the 4 heads of the $8 \times 8$ state matrix $S_t$, we visualize the 32 coordinates with the highest temporal variance.

Interpreting DeltaNet weights directly is challenging. We hence visualized the DeltaNet hidden states across several example data points, and found that portions of the hidden states are quite sensitive to periodic data, while some parts of the hidden state is used to track a gradual temporal evolution(see Figure 10, note the presence of both high frequency variations and low frequency graduations in response to periodic signals).

We posit that the effectiveness of our hybrid strategy is driven by convolutional filters learning more position-specific information, while the DeltaNet layer is learning more periodic information.

