# OpenReview forum: "Reverso: Efficient Time Series Foundation Models for Zero-shot Forecasting"
_ICML.cc/2026/Workshop/FMSD — FMSD @ ICML 2026 Poster_

### Official Review · Reviewer_bpuz · 2026-05-14
**Strong empirical results, but framing and rigor need tightening**

**Rating:** 7
**Confidence:** 4

**Review:**

## Summary

The paper proposes Reverso, a parameter-efficient foundation model architecture for
zero-shot univariate point forecasting. The architecture has three main components:
a multi-scale input strategy that creates downsampled views of the context, a stack
of hybrid blocks combining long convolutions, DeltaNet layers, and MLPs, and a
single attention-based decoder head. The authors train three versions of Reverso
at 0.2M, 0.6M, and 2.6M parameters. All variants significantly outperform baselines
of comparable size on zero-shot forecasting benchmarks.

## Strengths

1. Strong performance per parameter. Reverso significantly outperforms baselines
   of similar size and is competitive with models one to three orders of magnitude
   larger on Gift-Eval and LTSF benchmarks.
2. Architectural design. The paper combines several ideas, namely long
   convolutions, DeltaNet, attention-based decoding, multi-scale inputs, and
   inference-time flip equivariance, into a coherent and parameter-efficient
   recipe.
3. Broad baseline coverage. The paper compares against more than a dozen
   recent TSFMs across multiple benchmarks.
4. Efficiency analysis. Figure 3 reports both inference latency and peak memory
   usage against major baselines, supporting the efficiency claims.

## Areas for improvements

### 1. The main paper underrepresents results that the abstract advertises.

The abstract claims competitive performance on classification, anomaly detection,
and imputation, but the main body discusses only zero-shot forecasting. The
non-forecasting results are deferred entirely to Appendix C. Given that these tasks
are core to the abstract's framing, at least a summary table and discussion belong
in the main paper.

### 2. The anomaly detection methodology should be discussed more critically.

The model produces only point forecasts, and Appendix C.3 uses forecast MAE as the
anomaly score. This approach should be discussed with respect to its limitations.

### 3. The role of flip equivariance is under-discussed in the main paper.

Flip equivariance is introduced only in the appendix, but appears in Figure 3 of
the main paper. It is not stated whether the Reverso numbers in Table 2 and
Figure 2 were measured with flip equivariance or not.

### 4. The deployment story is not validated in deployment-relevant settings.

The abstract motivates the work by edge and on-device deployment, but all latency
and memory measurements are taken on an H100 GPU. To support the on-device claim,
benchmarks on CPU or mobile/edge hardware would be far more informative than H100 numbers, and would also better demonstrate the practical advantage of a 2.6M parameter model over models with 100M+ parameters.

### 5. The efficiency frontier comparison could be more complete.

Figure 3 suggests that Chronos-2 and TiRex may achieve favorable inference
efficiency per parameter. Including Chronos-2-Small (28M parameters) or scaling Reverso further (to ~10M–100M parameters) would clarify whether the efficiency advantage is a property of the architecture or specifically of the small-model regime.


## Justification of Score

The paper presents a coherent and well-engineered architecture that achieves
strong performance per parameter and would foster valuable discussion about
parameter efficiency in time series foundation models. The main contributions —
the multi-scale input strategy and the conv+DeltaNet hybrid — are well-motivated,
and the empirical results are convincing within their scope. However, the paper's
framing overreaches in places (zero-shot scope, deployment claims, headline
comparisons that depend on flip equivariance), and tightening the rigor on these points would substantially strengthen the camera-ready version.

---

### Official Review · Reviewer_Yd18 · 2026-05-21
**-**

**Rating:** 7
**Confidence:** 4

**Review:**

### Summary
Reverso is a family of small TSFMs (0.2M–2.6M parameters) for zero-shot forecasting. The architecture combines a multi-scale downsampled input, with a hybrid long-convolution + DeltaNet architecture. For training a set of established methods (sythetic data generation, augmentation, flip-equivariance etc.) are combined. Across multiple forecasting benchmark as well as on other downstream tasks (classification, anomaly detection, imputation) Reverse stays competitive with larger and slower TSFMs.

### Strengths
- Pushing TSFMs toward the small end of the parameter axis — and showing that a fast 2.6M-parameter model can stay competitive on Gift-Eval is a  useful and relevant direction. The deployment motivation (edge / on-device / memory-constrained inference) is concrete.
- The empirical evaluation is broad. Beyond zero-shot point forecasting on multiple established benchmark, the paper covers classification, anomaly detection and imputation. This kind of cross-task coverage in a workshop submission is unusual and should be positively highlighted.
- The paper reports four full training runs across seeds to show the consistency of results.

### Areas for Improvement
- §C.5 evaluates probabilistic forecasting by retraining the decoder head with a quantile-regression objective on top of a pretrained Reverso backbone. This means the Reverso model evaluated in the rest of the paper — the one with the headline Gift-Eval MASE of 0.706 — does not provide any probabilistic forecasting capability. Probabilistic  output is a standard and widely needed capability in forecasting. This is a big limitation worth noting in the Discussion
- Across §2.1 and §B.2, the paper makes a series of "we found X works better than Y" claims — hybrid LongConv+DeltaNet vs. either component alone vs. GLA / GDN / GDP variants; attention decoder vs. linea decoder; state-weaving on/off; [0,1]-norm vs. z-score; MLP vs. GLU channel-mixing; inference-time downsampling on/off — but no ablation table appears in the manuscript. We acknowledge that within a this submission, there is no way to ablate every choice at every model scale for every benchmark. However, a single consolidated ablation table on Gift-Eval at one model scale, covering the highest-leverage design choices  would already resolve this concern.

### Detailed Comments
Please clarify  if the inference latency and peak-memory numbers in Figure 3 measured with or without the flip-equivariance?

### Justification of Score
Reverso targets a useful research direction (small TSFMs), executes it carefully (chooses suitable architecture and combines it smartly with a bag of established training methods), and reports an unusually broad empirical evaluation.

---

### Official Review · Reviewer_CwWX · 2026-05-22

**Rating:** 7
**Confidence:** 4

**Review:**

## Summary
The paper builds a small TSFM, obtaining ponit forecasting results rivaling models >10x its size. They make use of recent advances in linear attention, specifically the DeltaNet architecture, combined with long convolution (sped up with FFT-based kernels). They also introduce data augmentation strategies for training and inference.

## Strengths
- Discussion section clearly highlights limitations and directions for future work
- Great results for a small model
- Gives further details in appendix

## Areas for Improvement
- Multi-scale input description is unclear. Formal notation or a diagram would be good to illustrate this point. I did not fully understand the method from the exposition.
- Decoder head exposition is also very unclear. I am unable to understand the exact architecture from the current writing. Clearer mathematical notation should be provided.
- Flip should be mentioned in the main paper given that it is presented in Figure 3. Do the main results use flip?
- Would be better to include ablation results in main section.

## Justification of score
The paper presents an interesting model with strong performance, showing that small models can achieve strong results on general forecasting benchmarks.